Different pruning level effects on flowering period and chlorophyll fluorescence parameters of Loropetalum chinense var. rubrum

Zhang Damao 1 2 3
Cai Wenqi 1 2 3
Zhang Xia 1 2 3
Li Weidong 4
Zhou Yi 1
Chen Yaqian 1
Mi Qiulin 1
Jin Lanting 5
Xu Lu 1 2 3
Yu Xiaoying 475705701@qq.com 1 2 3
Li Yanlin liyanlin@hunau.edu.cn 1 2 3
1 Hunan Agricultural University, College of Horticulture , Changsha , Hunan , China
2 Engineering Research Center for Horticultural Crop Germplasm Creation and New Variety Breeding, Ministry of Education , Changsha , China
3 Hunan Mid-Subtropical Quality Plant Breeding and Utilization Engineering Technology Research Center , Changsha , China
4 Hunan Key Laboratory of Innovation and Comprehensive Utilization , Changsha , China
5 Hunan Agricultural University, College of Oriental Science & Technology , Changsha , China
Dąbrowski Piotr
Electronic publication date: 2022 May 10
Publication date: 2022
Volume: 10
Electronic Location ID: e13406
Received 2022 Jan 3; Accepted 2022 Apr 18
Copyright: ©2022 Zhang et al.
Copyright year: 2022
Copyright holder: Zhang et al.
License: This is an open access article distributed under the terms of the Creative Commons Attribution License, which permits unrestricted use, distribution, reproduction and adaptation in any medium and for any purpose provided that it is properly attributed. For attribution, the original author(s), title, publication source (PeerJ) and either DOI or URL of the article must be cited.
License URL: https://creativecommons.org/licenses/by/4.0/

Keywords: Flowering period, Chlorophyll fluorescence, Prune, Woody-Ornamental-Plants, Loropetalum chinense var. rubrum

Funding: Key Laboratory of Hunan Province for Innovation and Comprehensive Utilization of Garden Flower Germplasm 2019TP1033 Innovation training program for college students of Hunan Agricultural University XCX2021044 National Innovation and Entrepreneurship Training Program for College Students 202112653017X Hunan Agriculture Establishment, Research and Cultivation of Universities’ Shuang Yi Current’ SYL201802026 SYL2019012 Open Project of Horticulture Discipline of Hunan Agricultural University 2021YYXK001 National Key Research and Development Program “Research on Key Technologies of Rural Ecological Landscape Construction” Sub-project “Research on Rural Plant Landscape Construction and Application Technology” 2019YFD1100404 The Forestry Science and Technology Innovation Foundation of Hunan Province for Distinguished Young Scholarship XLKJ202205 Innovation and Entrepreneurship Training Program of Hunan Province for College Students 201941937227 The Found of Changsha Municipal Science and Technology Bureau KQ2202227 The work is funded by the Key Laboratory of Hunan Province for Innovation and Comprehensive Utilization of Garden Flower Germplasm (2019TP1033), the Innovation training program for college students of Hunan Agricultural University (XCX2021044), the National Innovation and Entrepreneurship Training Program for College Students (202112653017X), the Hunan Agriculture Establishment, Research and Cultivation of Universities’ Shuang Yi Current’ (Grant No. SYL201802026 and SYL2019012), the Open Project of Horticulture Discipline of Hunan Agricultural University (2021YYXK001), the National Key Research and Development Program “Research on Key Technologies of Rural Ecological Landscape Construction” Sub-project “Research on Rural Plant Landscape Construction and Application Technology” (2019YFD1100404), The Forestry Science and Technology Innovation Foundation of Hunan Province for Distinguished Young Scholarship (XLKJ202205), the Innovation and Entrepreneurship Training Program of Hunan Province for College Students (201941937227), and The Found of Changsha Municipal Science and Technology Bureau (KQ2202227). The funders had no role in study design, data collection and analysis, decision to publish, or preparation of the manuscript.

==============================
“Pruning” is a simple and efficient way to control the flowering period, but it is rarely used in perennial woody ornamental plants. In this paper, Loropetalum chinense var. rubrum was pruned in different degrees, and the relationship between pruning intensity and flowering number, and flowering time and chlorophyll fluorescence parameters were compared. After statistics, it was found that pruning could advance blossoms of L. chinense var. rubrum; also, light and heavy cutting could both obtain a larger number of flowers. In addition, through correlation analysis, it was found that during the flowering period, the Rfd parameter of the unpruned treatment had a very significant positive correlation with the number of flowers FN, which was 0.81. In other pruning treatment groups, Rfd and FN also presented a certain positive correlation, indicating that the Rfd parameter can be used to predict the number of flowers during the flowering process of L. chinense var. rubrum. The research results provided a new idea for the regulation of the flowering period of L. chinense var. rubrum and other woody ornamental plants and laid the foundation for the diversified application of L. chinense var. rubrum.

Introduction

The flowering of plants is an important manifestation of their reproductive growth. Except for normal metabolism and dry matter accumulation during vegetative growth, the flowering process was also affected by other factors (Li & Zhang, 2012; Wang et al., 2017), such as changes in ambient temperature (Hendry et al., 2021), length of light (Kikuchi & Handa, 2009; Song, Ito & Imaizumi, 2013), the wavelength of light and the intensity of light, changes in the content of endogenous and exogenous hormones, etc. As a simple agronomic measure, “pruning” is widely used in fruit trees and vegetables to promote the formation of branches, stimulate growth, and promote flowering and fruit setting (Huang et al., 2020), to achieve the purpose of increasing production.

Among the ornamental plants, the main purpose of the florescence control measures is to obtain a large number of flowers quickly and stably. Therefore, pruning is mostly applied to herbaceous ornamental plants and rarely applied to perennial woody ornamental plants (Calatayud et al., 2008). Some studies have shown that jasmine under moderate pruning intensity can significantly increase the number of its flowers and force its flowering (Chopde et al., 2017). In cut roses, pruning of the twig at the base of 10 cm cannot only shorten the time of flowering but also increase the quality; it is also the most suitable for the rose of Corolla cut flowers (Zhang et al., 2018). After pruning the top of the ‘Nadorcott’ mandarin, the number of flowers increased significantly in the second year (Mesejo et al., 2020). Pruning and topping can better promote the differentiation of flower buds of old plum trees such as ‘Beauty Plum’ and regulate flowering (Zhang et al., 2020). Pruning and removing the leaves and petioles on some branches can significantly increase the expression of the LFY gene in the leaves of sugar apples (Annona squamosa L.) and induce flowering (Mo et al., 2015).

After pruning, the branches are reduced, the photosynthetic capacity of the leaves of the unpruned branches is increased to compensate for the decrease in photosynthesis of the leaves of the pruned parts (Desotgiu et al., 2012). At the same time, more light is obtained in the canopy. The quantum yield of the plant and the chlorophyll a and chlorophyll b are increased, and the photosynthetic efficiency is increased (Saifuddin et al., 2010; Turnbull, Adams & Warren, 2007), accumulation rate of organic matter increases. Under different pruning levels, the photosynthetic rate and transpiration rate of light and heavy pruning is higher than those of control and moderate pruning (Ma et al., 2020). Through the combination of inversion algorithm research and examples, it was found that the photosynthesis parameters are consistent with the changing trend of the maximum photochemical efficiency of PS II, indicating that the chlorophyll fluorescence parameters are closely related to photosynthesis (Qiu et al., 2015). Other studies have shown that at the beginning of flowering, the nitrogen content per leaf area of leaves near the inflorescence decreases, which leads to the decrease of photosynthetic electron transfer rate and the decrease of photosynthetic capacity (Urban et al., 2008).

Loropetalum chinense var. rubrum is a famous native ornamental plant in Hunan Province (Li et al., 2011a). This species was famous for its bright leaf color, while its flowers are also colorful and charming (Wang et al., 2020; Zhang et al., 2021). However, due to inconsistent flowering time between individuals, the variated flower color has not been better utilized (Li et al., 2011b). Therefore, using simple operation, outstanding effects, cost-controllable pruning, and other measures, the flowering process of L. chinense var. rubrum can be controlled and solve problems that are difficult to use the L. chinense var. rubrum; meanwhile, it can also break the limitation of the single-use mode of it and lay the foundation for the development of L. chinense var. rubrum to potting. It is of great significance to promote the diversified and sustainable development of the L. chinense var. rubrum.

Materials & Methods

Plant material and experimental layout

This experiment was carried out in the flower base of Hunan Agricultural University. The experimental materials included Loropetalum chinense var. rubrum and new Loropetalum chinense cultivars, ‘Da Yehong’, ‘Hei Zhengzhu’, and ‘Xiangnong Xiangyun’, all experimental materials are the progeny of cuttings that have been cultivated for two years with stable genetic background. Among them, ‘Da Yehong’ and ‘Hei Zhenzhu’ are the main products of Loropetalum chinense var. rubrum, the variety authorization number of ”Da Yehong” is S-SV-LCR-010-2003. ‘Xiangnong Xiangyun’ is a new cultivar of Loropetalum chinense (The breeding institution is Hunan Agricultural University and the cultivar authorization number is 20201102, Breed Granting Institution: National Forestry and Grassland Administration, China.). The materials were potted on September 3, 2020, and the cultivation medium (garden soil: organic fertilizer = 2:1, the physical and chemical conditions of the garden soil are as follows: pH was 4.4, organic matter content was 1.07%, total nitrogen content was 0.12%, and available potassium content was 54.64 mg/kg; the physical and chemical properties of the organic fertilizer are: inorganic nutrients account for 15% (N:P: K = 9:3:3), amino acids account for 1.5%, organic matter account for 35%, pH is 6.9, and water content is 26%) was unified, the size of containers is 175 mm * 138 mm * 185 mm, and the volume of each pot is 5.68L. When planting, 4.5L of soil was added to keep every experimental material normal growth. After potting, they were all placed in the greenhouse, the room temperature was controlled at 25 °C. The plants were irrigated every three days to keep the potting soil moist from causing drought stress, at the same time, ensure that the fertilization for each plant is consistent.

At the end of September 2020, 24 pots of plants with robust and consistent growth in each cultivar were selected and placed evenly on the seedbed in different areas. After confirmation, all the experimental plants were determined to have no flower bud differentiation. Each treatment was divided into 4 treatments and each treatment had six pots. Process A: trim 1/5; process B: trim 1/2; process C: trim with cap; process D: no trimming (CK), respectively numbered A1, A2, A3, A4, A5, A 6; B1, B2, B3, B4, B5, B6; C1, C2, C3, C4, C5, C6 and D1, D2, D3, D4, D5, D6. The pruning objects were chosen to be branches that have grown for two quarters in the current year, as shown in Fig. 1. After the trimming was completed, the materials were sorted and placed, then they would be maintained at room temperature at 25 °C, and the fertilizer and water management would be unified.

Figure 1 Schematic diagram of trimming position.

0∼4 is the length of the branches to be pruned; 0∼1 is 1/5 the pruning length; 0∼2 is 1/2 the pruning length; 0∼3 is the pruning length with cap (boundary between current season’s branches and previous season’s branches).

Measurements

After pruning, the growth of the material was observed every day. When the branch germinated, the number of buds (pieces/plant), the length of new shoots (cm), the number of flower buds (pieces/plant), and the number of flowers (flowers/plant) were recorded. Chlorophyll fluorescence parameters (initial fluorescence, maximum fluorescence, etc.) of each group were measured before pruning, after budding, and after flowering using a hand-held chlorophyll fluorometer (FluorPen FP 110). Before measuring, it is necessary to use a leaf clip to mark the leaves and acclimate them to the dark for 30 min. Then turn on the chlorophyll fluorometer and measure sequentially using the built-in measurement protocols “OJIP”, “NPQ3” and “QY”. After the measurement, the dedicated software (FluorPen 1.1.0.3) was used to export the data to a computer for subsequent analysis.

Statistical analysis

The experimental data were statistically analyzed using SPAS 22.0, the Duncan method was used for multiple comparisons, Origin 2019b was used for graph drawing, and the correlation analysis was performed in RStudio (RStudio 2019.09.0 Build 351; RStudio Team, 2019). Then JIP-test (Chlorophyll Fluorescence Induction Kinetic Curve Analysis Technology) was used to analyze the relationship between the changes of chlorophyll fluorescence parameters and the flowering process of the three test materials under different tests treatments.

Results

Effect of different pruning levels on flowering time

It was found that on the 20th day after the start of the experiment, all three experimental materials bloomed. Among them, the A treatment of ‘Xiangnong Xiangyun’ began to flower on the 14th day after the start of the pruning treatments, the B and C treatments began to flower on the 20th day, and the CK treatment began to flower on the 23rd day (Table 1). The flowering time of all pruned treatments have been advanced, and there was a significant difference in the unpruned experimental groups, ‘Hei Zhenzhu’ only flowered in the C and CK groups during the entire experimental period, and they flowered on the 20th to 23rd day after pruned (Table 1). While the flowering time of the pruning treatment groups is significantly different from it in the unpruned group; ‘Da Yehong’ only bloomed in group B and group C during the entire experimental period, and both bloomed on the 20th day of the experiment (Table 1). Compared with the unpruned groups, it is proved that pruning can promote the flowering of ‘Da Yehong’.

Table 1 The first flowering time of the three materials after pruning (unit: day).

	Hei Zhenzhu	Xiangnong Xiangyun	Da Yehong	
A (1/5 trimming)	0.00 ± 0.00a	13.80 ± 0.84a	0.00 ± 0.00a	
B (1/2 trimming)	0.00 ± 0.00a	19.60 ± 0.89b	19.40 ± 0.89b	
C (trim with cap)	19.80 ± 0.84b	19.60 ± 0.55b	19.60 ± 0.55b	
CK (no trimming)	22.60 ± 0.55bc	22.60 ± 0.55bc	0.00 ± 0.00a	
Notes.

Mean (±SE) with different letters are significantly different within the first flowering (mean separation by LSD and Duncan’s test at p < 0.05); ‘0’ means that the treatment did not bloom during the entire study period.

For these three kinds of materials, the A treatment can make ‘Xiangnong Xiangyun’ bloom 9 days in advance, the B treatment can make it bloom 3 days in advance, and the C treatment can make ‘Xiangnong Xiangyun’ and ‘Hei Zhenzhu’ bloomed 3 days in advance. In short, pruning can promote their flowering within a certain range.

Effect of different pruning levels on the number of flowers

The B treatment had the most flowers in total, but the difference between the maximum and the minimum was evident, which represented a large dispersion, and the number of flowers in each group was quite different; The A treatment ranked the second in the total number of flowering, and the extremum and the dispersion was small; the C treatment ranked third in the total number of flowering, and the extremum and dispersion are similar to the A treatment; the CK treatment had the least number of flowers (Fig. 2), indicating that a certain degree of pruning can increase the number of flowers.

Figure 2 Relationship between pruning level and number of flowerings.

‘A’ represents 1/5 trimming group, ‘B’ represents 1/2 trimming group, ‘C’ represents ‘trim with cap’ group, ‘CK’ represents no trimming group; The bottom edge of the box plots of B, C, and CK coincide with the quartile, instead of no bottom edge.

JIP-test curve during anthesis

Under the A group, the JIP-test curve of ‘Hei Zhenzhu’ and ‘Da Yehong’ showed the same change trend as the CK group at the early, peak and final flowering stages (Fig. 3); that is, the positions of the four points O, J, I, and P in the JIP-test curve all show that the full blooming period is lower than the initial blooming period and the final blooming period. In the A treatment of ‘Xiangnong Xiangyun’, the value of each point in the initial flowering period is the smallest, and the value of each point in the final flowering period is the largest, and the full flowering period lies between the two. However, in the CK group of ‘Xiangnong Xiangyun’, the value of each point in the initial flowering period is the largest, the value of each point in the full flowering period is the smallest, and the final flowering period is between the two. Calculated by the formula Fv = Fp − Fo. Under the A group, the Fv values of ‘Hei Zhenzhu’ are 38,157, 37,149, 44,280 at the beginning, full, and last flowering stages, the Fv value of ‘Xiangnong Xiangyun’ is 38,688, 40,704, 47,650 in the first, full, and last flowering stages, the Fv value of ‘Da Yehong’ is 41,679, 38,547, 46,891 in the first, full and last flowering period; Under CK group, the Fv value of ‘Hei Zhenzhu’ is 36,867, 35,113, 40,493 in the first, full, and last flowering stages, the Fv value of ‘Xiangnong Xiangyun’ is 51,204, 45,245, 47,206 in the first, full, and last flowering stages, the Fv value of ‘Da Yehong’ is 42,752, 39,674, 46,165 in the first, full, and last flowering periods.

Figure 3 Comparison of JIP-test curves of three experimental materials under treatment A and CK group during flowering period.

‘A’ represents 1/5 trimming group, ‘CK’ represents no trimming group; ‘L1’ represents ‘Hei Zhenzhu’, ‘L2’ represents ‘Xiangnong Xiangyun’, and ‘L3’ represents ‘Da Yehong’; ‘_C’ represents the first flowering stage, which refers to the situation where the plant has flower buds or flower buds and a few flowers, ‘_S’ represents the peak flowering stage, which refers to the situation where most or all of the flower buds are open and the number of flowers is large , and ‘_M’ represents the last flowering stage, which refers to the situation where all the flower buds are open, the flowers are wilted and the number of flowers is small; ‘O’ represents the initial fluorescence Fo, ‘L’ and ‘K’ represent the fluorescence amount at 150 us and 300 us respectively (only applicable under specific stress conditions), ‘J’ represents the fluorescence amount at 2 ms, ‘I’ represents the fluorescence amount at 30 ms, and ‘P’ represents the maximum fluorescence; and there is no unit for chlorophyll fluorescence parameters.

Under treatments B and C, the JIP-test curves of the three experimental materials ‘Hei Zhenzhu’, ‘Xiangnong Xiangyun’, and ‘Da Yehong’ showed the same changing trends in the initial flowering, full flowering, and final flowering periods (Fig. 4). That is, the positions of O, J, I, and P in the JIP-test curve all show a trend that the initial flowering period is lower than the full flowering period and the final flowering period. Among them, the fluorescence values of the three points J, I, and P of ‘Hei Zhenzhu’ are all lower than the corresponding values of ‘Xiangnong Xiangyun’ and ‘Da Yehong’. Undertreatment B, the Fv values of ‘Hei Zhenzhu’ are 34,874, 38,829, 42,231 at the beginning, full, and last flowering stages, the Fv value of ‘Xiangnong Xiangyun’ is 39,122, 41,982, 46,816 in the first, full, and last flowering stages, the Fv value of ‘Da Yehong’ is 42,447, 45,028, 47,650 in the first, full and last flowering period; Under C treatment, the Fv value of ‘Hei Zhenzhu’ is 35,101, 37,323, 43,370 in the first, full, and last flowering stages. The Fv value of ‘Xiangnong Xiangyun’ is 40,368, 43,858, 48,561 in the first, full, and last flowering stages. The Fv value of ‘Da Yehong’ is 40,433, 46,740, 51,790 in the first, full, and last flowering periods.

Figure 4 Comparison of JIP-test curves of three experimental materials under treatment B, C and CK group during flowering period.

‘B’ represents 1/2 trimming group, ‘C’ represents ‘trim with cap’ group, ‘CK’ represents no trimming group; ‘L1’ represents ‘Hei Zhenzhu’, ‘L2’ represents ‘Xiangnong Xiangyun’, and ‘L3’ represents ‘Da Yehong’; ‘_C’ represents the first flowering stage, which refers to the situation where the plant has flower buds or flower buds and a few flowers, ‘_S’ represents the peak flowering stage, which refers to the situation where most or all of the flower buds are open and the number of flowers is large, and ‘_M’ represents the last flowering stage, which refers to the situation where all the flower buds are open, the flowers are wilted and the number of flowers is small; ‘O’ represents the initial fluorescence Fo, ‘L’ and ‘K’ represent the fluorescence amount at 150 us and 300 us respectively (only applicable under specific stress conditions), ‘J’ represents the fluorescence amount at 2 ms, ‘I’ represents the fluorescence amount at 30 ms, and ‘P’ represents the maximum fluorescence; and there is no unit for chlorophyll fluorescence parameters.

Correlation analysis of flowering number and chlorophyll fluorescence parameters

Through correlation analysis, it was found that the number of flowers FN in the CK group during the flowering period was extremely significantly positively correlated with the chlorophyll fluorescence parameter Rfd, and the correlation coefficient was 0.81 (Fig. 5). At the same time, we found that in the A, B, and C pruning treatment groups, FN and Rfd are all positively correlated, with correlation coefficients of 0.30, 0.50, and 0.49, with an average of 0.43. In the treatment groups A and B, the chlorophyll fluorescence parameters PS II (Fv/Fm) and QY_max were negatively correlated with FN, and the correlation coefficients were −0.41, −0.47, −0.45, −0.55, However, in the C and CK groups, PS II and QY_max were positively correlated with FN, and the correlation coefficients were 0.44, 0.29, 0.31, and 0.11.

Figure 5 Correlation analysis of flowering number and chlorophyll fluorescence parameters.

‘A’ represents 1/5 trimming group, ‘B’ represents 1/2 trimming group, ‘C’ represents ‘trim with cap’ group, ‘CK’ represents no trimming group; ‘FN’ represents the number of flowers during the flowering period; ‘ Fo’ represents the initial fluorescence; ‘ Fm’ represents the maximum fluorescence; PSII represents the efficiency of the photosystem II, the calculation formula is (Fm−Fo)/Fm; ‘ Rfd’ represents the Ratio of fluorescence decline, ‘QY_m’ represents the maximum quantum yield; and there is no unit for chlorophyll fluorescence parameters.

Discussion

The effect of pruning intensity on flowering time and number

As a simple agronomic measure, “pruning” is widely used in fruit and vegetable production. Pruning can promote the formation of branches and stimulate vegetative growth, thereby achieving the purpose of increasing yield (Jorquera-Fontena, Alberdi & Franck, 2014; Pineda et al., 2020). However, the application of this simple measure in the field of ornamental plants is not popular.

Through this experiment, we found that ‘1/2 trimming’ and ‘trim with cap’ can promote the flowering of L. chinense var. rubrum and L. chinense and advance the flowering period for 3 days. This result shows that pruning can destroy the balance between vegetative growth and reproductive growth during the normal growth of plants, As the pruning intensity increases, more dormant buds are activated, causing an increase in leaf area, to accelerate the accumulation of dry matter (Amarnath, Mishra & Singh, 2020; Huang et al., 2020; Persello et al., 2019; Weraduwage et al., 2015). At the same time, the distribution of endogenous hormones in the branches changes after pruning. The content of ZR and GA3 in the buds at the lower part of the cut is increased, and the content of IAA and ABA is reduced, thereby promoting the germination of the buds under the cut and the formation of flower buds (Booker, Chatfield & Leyser, 2003; Liu et al., 2021; Mai-Mai-Ti et al., 2013). The phenomenon of “1/5 trimming’ can advance the flowering date by 9 days’ in L. chinense may be due to its strong photosynthetic capacity and rapid accumulation of nutrients (Wang et al., 2020). Low-level pruning can quickly destroy the balance between vegetative growth and reproductive growth (Gaaliche et al., 2011; Kovaleski et al., 2015), thus it can stimulate the plant to bloom earlier.

Studies have shown that within 48 h after pruning, the sugar and starch of the buds on the plant stems will be temporarily lost, and then will increase rapidly and reach a stable level. The changes in nutrients directly promote the development of flower buds (Girault et al., 2010). So, through proper pruning, the yield of Lonicera japonica can be significantly increased (1.4 times) (Qin et al., 2019). After pruning, the carbohydrates of the “source” and “sink” of the plant are redistributed (Berman & Dejong, 2015; Dambreville et al., 2015), in this process, the accumulation of nutrients required for reproductive growth may have been satisfied. Combined with the box plot, after 1/2 level pruning, the balance between the number of dormant buds on the branches below the pruning incision and the energy stored by the branches that can supply flowering is ensured (Pala et al., 2014), so the maximum number of flowers can be obtained. However, the number of flowers varies greatly within the group, so it can be used as a measure to control flowers per plant. Although the number of flowers obtained by 1/5 level pruning is less than that of 1/2 level pruning, the difference in the number of flowers within the group is smaller, which means it is more suitable for the situation that requires a neater flower amount and flowering period, that is, overall control of flowers.

The relationship between the number of flowering and chlorophyll fluorescence parameters

After trimming, the speed of electron transfer during photosynthesis is accelerated, which promotes photosynthesis capacity and is more conducive to the accumulation of nutrients (Turnbull, Adams & Warren, 2007), and the canopy structure changes and the interior of the canopy have good air permeability, which is more conducive to the progress of photosynthesis (Forrester et al., 2012; Song et al., 2010). Combining Figs. 2 and 3, it is found that the initial fluorescence Fo and maximum fluorescence Fm of L. chinense is greater than that of the two types of L. chinense var. rubrum, which reflects a certain extent that the photosynthetic capacity of L. chinense is greater than that of the two types of L. chinense var. rubrum (You & Gong, 2012). The photosynthesis ability is strong, so it can accumulate the organic nutrients needed for flowering more quickly, which explained why it could bloom quickly after low-level pruning stimulation. In the photosynthetic system of plants, when the PSII supply-side plastoquinone QA decreases, it is manifested as the chlorophyll fluorescence rises from Fo to Fm (or Fp) (Banks, 2017). That is, Fv reflects the reduction of QA (Çiçek et al., 2019). Comparing the three experimental materials, it is found that the Fv value in the full flowering stage is less than the initial flowering stage and the final flowering stage, indicating that the photosynthetic system of these plants has a stronger reducing ability to plastoquinone QA in the full flowering stage. This is because, during the flowering process, flowers are a highly active reservoir, and need to obtain a large amount of water and soluble sugar from the source of the leaves (Yuan et al., 2016). To support the increasing demand for sugar during the flower opening, the photosynthetic activity of the leaves is greatly improved (Christiaens et al., 2015; FuJII & Kennedy, 1985).

The Rfd value in the chlorophyll fluorescence parameter is an effective indicator reflecting plant vitality and photosynthetic efficiency (Shin et al., 2017), in the early stage of short-term drought stress, the Rfd values of the two Arabidopsis mutants showed a downward trend and showed a significant difference, but Fv/Fm (PS II efficiency) did not show a significant difference (Yao et al., 2018), in the short term, it is more sensitive to the outside world than Fv/Fm (Lysenko et al., 2014). However, the changes of Fv/Fm and Rfd tend to be consistent during the long-term response (Sun et al., 2019). Other studies have shown that when the tomato fruit grows vigorously in the green ripe period, its Rfd value is significantly higher than the value of the ripe fruit (Abdelhamid et al., 2021). Therefore, in this experiment, the number of flowers during flowering was positively correlated with the Rfd value, especially in the CK group, there was a very significant positive correlation between Rfd and the number of flowers. In the treatment groups A and B, the higher the degree of pruning, the greater the negative correlation between Fv/Fm and QY_max and the number of flowers. Because the “source–sink” balance on the branches is broken as the degree of pruning increases on the branches of the season, the vegetative growth of a large number of latent buds as “sinks” is activated (Chen, 2019). At the same time, the balance between reproductive growth and vegetative growth is broken, which promotes reproductive growth (Salih et al., 2021). Some studies believe that in the process of flower induction, the branches of the pruning treatment site cannot undergo photosynthesis in a short period (Mo et al., 2015), thus causing this phenomenon. However, we speculate that when the source of the branches of the season is damaged, the undamaged “source” will improve photosynthetic capacity, while the sprouting flowers and many buds are at a disadvantage in the competition for nutrient distribution. We only paid attention to the relationship between the number of blooms and the undamaged “source”, which led to the occurrence of groups A and B. In the C group, since the branches of the current season were directly removed, there was no damage to the “source” part in the branches of the previous season. Therefore, the trend of the related data of this group and the CK group was consistent.

Conclusions

In summary, pruning can promote flowering by breaking the balance between vegetative growth and reproductive growth in plants. Through our research, we found that chlorophyll fluorescence parameters can not only be used as a non-destructive monitoring method for fruit quality (Lou et al., 2012), but also a potential method for non-destructive monitoring of flowering. A large number of studies have shown that plants will accelerate their nutrient accumulation speed when they encounter adversity stress (Kazan & Lyons, 2015; Zhang et al., 2016), leading to early flowering and fruiting and completing their life cycle as soon as possible (Ma et al., 2015); thus, we can regard the “pruning” measure as a kind of adversity stress to the plant, and the plant can bloom in advance by sensing this adversity and accelerate the completion of its life cycle. This issue is worthy of in-depth consideration.

Supplemental Information

File S1 Raw data for Table 1 and Figure 2

Click here for additional data file.

File S2 Raw data for Figure 3 and Figure 4

Click here for additional data file.

File S3 Raw data for Figure 5

Click here for additional data file.

File S4 SPSS data and SPSS original analysis results used for the relationship between pruning intensity and initial time

Click here for additional data file.

File S5 Comparison of phenotypes between the experimental group and the control group of each experimental material at the first flowering stage

Click here for additional data file.

File S6 Schematic diagram of pruning or comparison diagram before and after pruning

Click here for additional data file.

Additional Information and Declarations

Competing Interests

Author Contributions

Data Availability

The authors declare there are no competing interests.

Damao Zhang conceived and designed the experiments, performed the experiments, analyzed the data, prepared figures and/or tables, authored or reviewed drafts of the paper, and approved the final draft.

Wenqi Cai performed the experiments, analyzed the data, authored or reviewed drafts of the paper, and approved the final draft.

Xia Zhang analyzed the data, authored or reviewed drafts of the paper, and approved the final draft.

Weidong Li and Lu Xu conceived and designed the experiments, authored or reviewed drafts of the paper, and approved the final draft.

Yi Zhou and Lanting Jin analyzed the data, prepared figures and/or tables, and approved the final draft.

Yaqian Chen and Qiulin Mi performed the experiments, prepared figures and/or tables, and approved the final draft.

Xiaoying Yu and Yanlin Li conceived and designed the experiments, prepared figures and/or tables, authored or reviewed drafts of the paper, and approved the final draft.

The following information was supplied regarding data availability:

The raw data are available in the Supplementary Files.

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
