# Peer review of "Different pruning level effects on flowering period and chlorophyll fluorescence parameters of Loropetalum chinense var. rubrum"

_PeerJ, doi:10.7717/peerj.13406_

## Round 0.1 · original submission · Major Revisions

Noble Authors,

Two independent experts assessed your work. Both commented that it could be published in PeerJ, but must be corrected beforehand. I kindly ask you to read the comments of both reviewers carefully and to respond to them all.

With best regards,

Reviewer 1 ·

Basic reporting

Level of langauge is acptable and literature reference is sufficiend to describe problem which presented by authors

Experimental design

There is some problem in part of materials and method. Detailed comments and suggestions were added in atached file

Validity of the findings

Article is interesting and quite good and will be better with added suggested information

Additional comments

Interesting work for those who like ornamental plants

Review of peerj-69279 manuscript – suggestion for authors (from the attached PDF)
In part Materials & Methods Plant material and experimental layout authors only how the plant
were look before trimming process but they do not show it after it. There is also lack of
information how the plant material was protected for drying process. In line 93-94 authors also
write they used home compost but they forget to put information about pH of it and some basic
chemical characteristic such C:N ratio content of N, C, P and K. Authors used in experiment also
organic fertilizer but they do not write what was it and it chemical characteristic. There is also
lack of information about size of pot which were used and how much medium was putted to each
pot. In section measurements authors write they measured chlorophyll fluorescence but they
didn’t write what was the fluorimeter they used and described the measurement protocol and
haven’t write information did they adapt leaves. If yes for how long.

Annotated reviews are not available for download in order to protect the identity of reviewers who chose to remain anonymous.

Reviewer 2 ·

Basic reporting

"Pruning" is a simple and efficient way to control the flowering period, in this paper, Loropetalum chinense var. rubrum was pruned in different degrees. This result has certain significance for regulating the flowering of woody ornamental plants, but there are some problems in the article that need to be revised and improved.
1. Materials and methods
1) The description of pruning is not clear, and the difference between the methods of various pruning degrees is not clearly indicated. It is recommended to add a schematic diagram of pruning method. In particular, it is necessary to explain the state of the plant before pruning, whether it has no flower bud or has a flower bud? And also, the age of the plant materials, the reasons for selecting these varieties etc. should be explained in detail. These will affect whether the conclusions of the experiment are generalizable.
2. In the results
1) The error value in Figure 2 is too large, it is recommended to increase the repetition to reduce the error.
2) Please check the description of the data in Table 1 carefully. First, no unit is added in the table. Second, does the 0 day in the table mean that the flower began to bloom after 0 days of pruning? This result is unconvincing.
3) There is no clear description of 'the early, peak and final flowering stages' in JIP-test curve during anthesis.

Experimental design

The materials and methods used should be described in detail, especially the choice of materials.

Validity of the findings

no comment

Additional comments

Pruning is very important for woody ornamental plants to control flowering and plant type. The results of this paper can provide a good theoretical support for the cultivation and management of woody ornamental plants. However, the data and detailed description need to be further deepened in the article, and it is also suggested to increase the flowering phenotype of the pruned plants, so that readers can intuitively have a perceptual understanding of the pruning results.

---

## Round 0.2 · accepted · Accept

Noble Authors,

Based on the reviews of your work and the corrections which You made, I decided to allow this article to be published.

With best regards,

Reviewer 1 ·

Basic reporting

English language is correct and literature references well choosen to describe problem which is presented in paper

Experimental design

All previous mention mistakes and problems which were noticed was corrected and eliminated by authors

Validity of the findings

Very interesting paper for producers of ornamental plants

Additional comments

You prepare very interesting paper which some scientific potential for people who are interested in subject of ornamental plants